# Nutritional Intervention Facilitates Food Intake after Epilepsy Surgery

**DOI:** 10.3390/brainsci11040514

**Published:** 2021-04-17

**Authors:** Rika Suzumura, Ayataka Fujimoto, Keishiro Sato, Shimpei Baba, Satoko Kubota, Sayuri Itoh, Isamu Shibamoto, Hideo Enoki, Tohru Okanishi

**Affiliations:** 1Department of Nutrition, Seirei Hamamatsu General Hospital, Shizuoka 430-8558, Japan; riccaricca0908@gmail.com (R.S.); s-kubota@sis.seirei.or.jp (S.K.); sasagawa@sis.seirei.or.jp (S.I.); 2Comprehensive Epilepsy Center, Seirei Hamamatsu General Hospital, Shizuoka 430-8558, Japan; k-sato@sis.seirei.or.jp (K.S.); sbaba@sis.seirei.or.jp (S.B.); enokih.neuropediatr@gmail.com (H.E.); t.okanishi@tottori-u.ac.jp (T.O.); 3Seirei Christopher University, Shizuoka 433-8558, Japan; isamu-s@seirei.ac.jp

**Keywords:** perioperative nutritional intervention, epilepsy surgery, total food intake, total infusion amount, nutritionist, infection

## Abstract

Background: We investigated whether nutritional intervention affected food intake after epilepsy surgery and if intravenous infusions were required in patients with epilepsy. We hypothesized that postoperative food intake would be increased by nutritional intervention. The purpose of this study was to compare postoperative food intake in the periods before and after nutritional intervention. Methods: Between September 2015 and October 2020, 124 epilepsy surgeries were performed. Of these, 65 patients who underwent subdural electrode placement followed by open cranial epilepsy surgery were studied. Postoperative total food intake, rate of maintenance of food intake, and total intravenous infusion were compared in the periods before and after nutritional intervention. Results: A total of 26 females and 39 males (age range 3–60, mean 27.1, standard deviation (SD) 14.3, median 26 years) were enrolled. Of these, 18 females and 23 males (3–60, mean 28.2, SD 15.1, median 26 years) were in the pre-nutritional intervention period group, and eight females and 16 males (5–51, mean 25.2, SD 12.9, median 26.5 years) were in the post-nutritional intervention period group. The post-nutritional intervention period group showed significantly higher food intake (*p* = 0.015) and lower total infusion (*p* = 0.006) than the pre-nutritional intervention period group. Conclusion: The nutritional intervention increased food intake and also reduced the total amount of intravenous infusion. To identify the cut-off day to cease the intervention and to evaluate whether the intervention can reduce the complication rate, a multicenter study with a large number of patients is warranted.

## 1. Introduction

Malnutrition is detrimental for postoperative patients [1] and is associated with a poor prognosis after trauma [2,3] and burns [4]. Proper nutrition is needed to facilitate postoperative wound healing [5]. Enteral, parenteral, and transdermal delivery systems are available for providing nutrition [6,7,8]. We generally use the parenteral route for patients who are unable to use the enteral route. Even though the parenteral delivery system has more advantages than the oral route in some situations [6], the enteral route is generally recommended due to the known complications of parenteral nutrition [9,10,11,12,13].

Patients with focal onset epilepsy sometimes require invasive monitoring prior to subsequent focal resection surgery. Adverse effects of invasive electrode surgery can include wound infection, meningitis, cerebrospinal fluid leak, hematoma, and brain edema [14,15,16,17]. Perioperative management to reduce these complications by reducing the contact number of electrodes and using both subdural grids and depth electrodes has been studied [18,19,20]. In addition, appropriate nutrition can reduce wound infection, increase immunity, and facilitate wound healing [21,22]. However, food intake following epilepsy surgery has not been widely studied. Therefore, whether nutritional intervention affected post-epilepsy surgery food intake was retrospectively investigated in this study. We hypothesized that postoperative food intake would be increased by nutritional intervention. The purpose of this study was to compare postoperative food intake between the periods before and after nutritional intervention as the primary outcome.

## 2. Methods

The ethics committee at Seirei Hamamatsu General Hospital approved the study protocol in accordance with the principles of the Declaration of Helsinki (approval number: 3515). This cross-sectional, observational, non-randomized study was carried out as a retrospective electronic chart review of patients treated between September 2015 and October 2020 in Seirei Hamamatsu General Hospital.

### 2.1. Clinical Information

We performed 124 intracranial electrode insertion surgeries for patients with epilepsy between September 2015 and October 2020. Among these, patients who fulfilled the following criteria were selected: (1) age ≥ 3 years old; (2) patients who received nutrition orally before the surgery; and (3) patients who underwent subdural electrode surgery (SE) followed by open cranial epilepsy surgery such as focal resection or disconnection surgery. In our facility, patients are admitted 1 day before SE, which is followed by focus resection, disconnection surgery, or sometimes just removal of the electrodes; they are then discharged. We introduced nutritional intervention in April 2018, and thus, we chose the term from September 2015 to October 2020 to compare each period of 30 months before and after establishment of nutritional intervention. Patients <3 years old were excluded from the study because they may receive inconsistent styles of nutrition, such as milk or various types of baby food. In addition, some of them were on tube feeding, and thus, nutritional intervention was not performed. Patients who fulfilled the following criteria were also excluded: (1) patients who received tube feeding; (2) patients who underwent depth electrode placement surgery through burr holes; (3) patients who underwent multiple SEs because the first SE did not cover an appropriate area such as the edge of the grid or a region outside the grid, and then they underwent second or third SEs; (4) patients who exhibited postoperative disturbance of consciousness due to status epilepticus, major complications such as massive subdural hematoma, severe meningitis, or other factors that prevented postoperative nutritional intervention and required prolonged infusion therapy; and (5) patients who were on a ketogenic diet including the classical ketogenic or modified Atkins diet. Patients who underwent burr hole surgery were excluded because this surgery is less invasive than open cranial electrode placement, and there could be differences in intracranial pressure [23]. Patients who underwent multiple SEs were excluded because we could not simply perform comparisons before and after the introduction of the nutritional intervention for any of these patients who exhibited postoperative disturbance of consciousness. The enrolled patients were divided into the pre-nutritional intervention period group (pre-N group) and the post-nutritional intervention period group (post-N group) according to the time of introduction of nutritional intervention in April 2018.

### 2.2. Nutritional Intervention

The daily caloric intake for each patient who underwent SE was calculated by nutritionists based on the Harris–Benedict equation [24]. The basic energy expenditure for females was 655.1 + (9.563 × Body weight) + (1.85 × Height) − (4.676 × Age), and that for males was 66.5 + (13.75 × Body weight) + (5.003 × Height) − (6.775 × Age). The result was multiplied by the coefficient 1.2 as the activity factor for the sedentary level (little or no exercise) because they underwent major surgery. If a patient was discharged within a week after the open cranial epilepsy surgery, the day of discharge was the last observed day. The nutritionists preoperatively explained to the patients that they could change their food form such as paste meal, as well as food volume. Postoperatively, the nutritionists evaluated the amount of food intake for each patient and changed the food form and volume without changing the total calories. This evaluation and food form and volume change were continued until the day of discharge.

### 2.3. Primary Outcome

#### 2.3.1. Observed Period for Food Intake

The period to evaluate food intake and adjust the food form and volume in this study was from the day of SE to 1 week after open cranial epilepsy surgery. One week after open cranial epilepsy surgery was selected instead of the day of discharge to ensure a uniform time period.

#### 2.3.2. Total Food Intake

The percentages of the total amount of intake per total expected basic expenditure during the observed periods were compared between the pre- and post-N groups.

#### 2.3.3. Time to Reach the Maximum Amount

The time required for patients to eat the maximum amount after open cranial epilepsy surgery was also compared between groups. The ratio of the maximum intake was calculated with the basic energy expenditure as 100, and this coefficient was divided by the number of days until maximum intake and used as the time. For example, if a patient reached 80% of the basic expenditure, which was the maximum amount in the post-open cranial epilepsy surgery period, and it took 4 days, the time was considered to be 20 (80%/4 days).

#### 2.3.4. Maintenance Rate

The ratio between the actual total intake from the date when the maximum intake was reached to the last day of observation after the open cranial epilepsy surgery and the estimated intake, assuming that the maximum intake continued until the last observed day, was calculated. For example, if a patient reached the maximum amount in the post-open cranial epilepsy surgery period on day 5, summation of the actual intake from day 5 to the last observed day (1866 kcal on day 5, 1400 kcal on day 6, and 1800 kcal on day 7) was divided by the estimated maximum intake, which was considered to continue until the last observed day (1866 kcal × 3). In this case, the ratio was (1866 + 1400 + 1880)/(1866 × 3). The ratios between the pre- and post-N groups were compared.

### 2.4. Secondary Outcome

The total number of days of admission, minor complications, and total volume of postoperative intravenous infusion were compared between the pre- and post-N groups.

### 2.5. Statistical Analysis

The Student’s *t*-test, Mann–Whitney U-test, and Fisher’s exact probability test were used, as appropriate. Statistical significance was set at *p* < 0.05. All analyses were conducted using Sigma plot (Systat Software, Inc., San Jose, CA, USA).

## 3. Results

### 3.1. Clinical Information

Of the 124 patients who underwent SEs, 65 met the criteria, including 26 females and 39 males with an age range from 3 to 60 (mean 27.1, standard deviation (SD) 14.3, median 26) years. Of these, 18 females and 23 males, ranging in age from 3 to 60 (mean 28.2, SD 15.1, median 26) years, were in the pre-N group, and eight females and 16 males, ranging in age from 5 to 51 (mean 25.2, SD 12.9, median 26.5) years, were in the post-N group. The number of days after SE, type of surgery, and total admission days are shown in Table 1. There were no significant differences in any of these parameters between the two groups.

### 3.2. Primary Outcome

#### 3.2.1. Total Food Intake

The total food intake ranged from 7% to 95% (mean 49%, SD 23%, median 49%). In the pre-N group, total food intake ranged from 7% to 86% (mean 43%, SD 21%, median 39%). In the post-N group, total food intake ranged from 20% to 95% (mean 58%, SD 24%, median 58%). The post-N group showed a significantly higher total food intake than the pre-N group (*p* = 0.015) (Table 2).

#### 3.2.2. Time to Reach the Maximum Amount

In the pre-N group, the time from 1 day after open cranial epilepsy surgery to the day that the patient reached the maximum intake ranged from 2.5% to 50%/day (mean 17.2, SD 11.1, median 14.3). In the post-N group, the time ranged from 9.2 to 100%/day (mean 27.3, SD 24.6, median 20.0). The pre-N group reached the maximum intake significantly faster than the post-N group (*p* = 0.040).

#### 3.2.3. Maintenance Rate

In the pre-N group, the maintenance rate from the day of the maximum intake to the last observed day ranged from 8% to 100% (mean 92%, SD 17%, median 100%). In the post-N group, the rate ranged from 68% to 100% (mean 97%, SD 9%, median 100%). No significant difference was observed between the two groups (*p* = 0.477).

### 3.3. Secondary Outcome

The number of days of admission in the pre-N group ranged from 10 to 49 (mean 21, SD 9.3, median 19) days, and for the post-N group, the number ranged from 10 to 29 (mean 18, SD, 4.8, median 17) days. No significant difference was found between the groups (*p* = 0.394). The total amount administered by intravenous infusion during the observation period in the pre-N group ranged from 5500 to 34,000 (mean 14,585, SD 6029, median 14,500) mL. The total amount in the post-N group ranged from 4000 to 28,000 (mean 10,833, SD 5782, median 9750) mL. The pre-N group required significantly more infusion than the post-N group (*p* = 0.006).

In the pre-N group, two wound infections occurred (5%) that were treated with oral antibiotics, and one patient had mild meningitis (2%) for 3 days with a moderate fever and slight neck stiffness. No complications were seen in the post-N group. There was no significant difference in minor complications between the groups (*p* = 0.290).

## 4. Discussion

Nutritional intervention increased the total amount of food intake after epilepsy surgery. The time to reach the maximum amount was faster in the pre-N group, but the maintenance rates in both groups were not different, meaning that postoperative appetite loss had continued in this study. Nutritional intervention did affect the amount of food intake postoperatively. The hypothesis that the amount of food intake increases following nutritional intervention was supported in this study.

Malnutrition can cause tissue and organ dysfunction and delay the wound healing process [25]. Thus, preventing malnutrition is important. Malnutrition can also weaken the immune system and lead to infection [26,27]. Additionally, immediate starting of oral or enteral feeding after surgery and shortening of the fasting period can reduce metabolic disturbance, enhance immunity, and improve tissue healing. These factors are considered to lead to early postoperative recovery [28,29,30].

The endocrine response as a defensive response to a number of factors, including surgical invasion, severe head trauma by catecholamines, cortical steroids, and glucagon, leads to stabilization and maintenance of the circulatory system and regulation of the immune system [31]. These hormonal responses require protein hypercatabolism [32,33]. Because excessive protein hypercatabolism is related to higher mortality and morbidity rates [27,34,35], enteral feeding immediately after surgery has beneficial effects [36,37].

Although nutritional intervention did not reduce the total number of days of admission, the intervention reduced the total amount of intravenous infusion in this study. We considered that the length of hospitalization was influenced by multiple factors, such as social background; therefore, no significant difference was observed. However, the intervention reduced the total amount of intravenous infusion, which could contribute to a reduction in medical costs and reduce the probability of iatrogenic complications such as heart failure [38] or electrolyte imbalance [39].

The number of patients in each group differed, even though the recruiting periods of both groups were 30 months each. This was probably due to the current trend of an increased tendency to perform depth electrode placement via burr holes, leading to an increase in the number of excluded cases in the post-N group.

One study limitation was that the appropriate timing for stopping nutritional intervention could not be determined. Once patients reached the maximum amount, nutritional intervention did not affect the amount of intake for the remaining days in this study. To identify the cut-off day to cease the intervention, more cases would need to be studied. Another study limitation was that, because of the small number of patients, the complication rate was not significantly different, even though the occurrence rate of complications was higher in the pre-N group than in the post-N group. Therefore, the benefit of nutritional intervention in this study appeared to only be a reduction in the total amount of infusion. For a clearer perspective of the effect of nutritional intervention, we plan to collect more data from multiple facilities.

## 5. Conclusions

Nutritional intervention increased the total amount of food intake after epilepsy surgery. The intervention also reduced the total amount of intravenous infusion. To identify the cut-off day to cease the intervention and to evaluate whether the intervention can reduce the complication rate, a multicenter study with a large number of patients is warranted.

## Figures and Tables

**Table 1 brainsci-11-00514-t001:** Clinical information.

	Pre-N Group	Post-N Group	*p*-Value
Age, y	mean 28.2, SD 15.1, median 26	mean 25.2, SD 12.9, median 26.5	*p* = 0.414
Sex	18 females, 23 males	8 females, 16 males	*p* = 0.409
Duration of admission, days	mean 20.9, SD 9.3, median 19.0	mean 17.19, SD 4.8, median 17.0	*p* = 0.394
Time after SE, days	mean 5.9, SD 2.5, median 6.0	mean 5.5, SD 1.6, median 6.0	*p* = 0.963
Minor complications	2 wound infections, 1 mild meningitis	0	*p* = 0.290
Surgery (pre-N group:post-N group)			*p* = 0.705
Temporal focus resection	20	13	
Frontal focus resection	13	7	
Occipital focus resection	1	1	
Corpus callosotomy	3	0	
Posterior quadrant disconnection	3	2	
Hemispherotomy	0	1	
Frontal + temporal focus resection	1	0	

N, nutritional intervention period; SD, standard deviation; SE, subdural electrode surgery.

**Table 2 brainsci-11-00514-t002:** Total food intake, time to maximum intake, maintenance rate, total number of days of admission, and required amount of intravenous infusion.

Outcomes	Pre-N Group	Post-N Group	*p*-Value
Total food intake	7 to 86% (mean 43, SD 21, median 39)	20 to 95% (mean 58, SD 24, median 58)	*p* = 0.015 *
Time to the maximum intake	2.5 to 50 (mean 17.2, SD 11.1, median 14.3)	9.2 to 100 (mean 27.3, SD 24.6, median 20.0)	*p* = 0.040 *
Maintenance rate	8 to 100% (mean 92, SD 17, median 100)	68 to 100% (mean 97, SD 9, median 100)	*p* = 0.477
Total number of days of admission	10 to 49 days (mean 21, SD 9.3, median 19)	10 to 29 days (mean 18, SD 4.8, median 17)	*p* = 0.394
Total infusion amount	5500 to 34,000 mL (mean 14,585, SD 6029, median 14,500)	4000 to 28,000 mL (mean 10,833, SD 5782, median 9750)	*p* = 0.006 *

N, nutrition intervention era; SD, standard deviation, * significant difference, *p* < 0.05.

## Data Availability

The data are not publicly available due to patients’ privacy.

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
