# Peer review of "Nutritional Intervention Facilitates Food Intake after Epilepsy Surgery"

_brainsci, 2021, doi:10.3390/brainsci11040514_

Round 1

Reviewer 1 Report

line 162

The pre-N group reached the maximum intake significantly faster than the pre-N group (p = 0.040). --> This part should be changed to the post-N group.

Table 2.

I think it will be easier to understand the contents by arranging the preN and postN groups as separate columns and comparing them as below.

Outcomes Pre-N Post-N P value
Total food intake      
Time to maximum intake      
Maintenance rate      
and so on...      

Author Response

Response to comments from reviewers

Reviewer #1

We greatly appreciate the thorough review and kind advice, which has helped us improve the quality of our manuscript. The manuscript has been revised to address the issues raised by the reviewers. The changes made and our point-by-point responses to the comments from the reviewer are summarized below. We have also tracked the changes in red in the revised manuscript.

#1. line 162

The pre-N group reached the maximum intake significantly faster than the pre-N group (p = 0.040). --> This part should be changed to the post-N group.

Response: Thank you very much for pointing out our mistake. We have corrected it as suggested.

#2. Table 2.

I think it will be easier to understand the contents by arranging the preN and postN groups as separate columns and comparing them as below.

Response: Thank you very much for this advice. We have revised Table 2 in accordance with the suggestion.

Reviewer 2 Report

This is an interesting study that investigated whether nutrition intervention affected food intake after epilepsy surgery and if intravenous infusion was required in patients with epilepsy. Overall, English language is fine; I suggest to check throughout the text for spelling errors and consistent use of abbreviations. The Methods section is clear and provides useful information; inclusion and exclusion criteria, length of follow up, and ethical committee approval have been stated in the text. Primary and secondary outcomes are described adequately. Data in Tables 1 and 2 are quite difficult to read; perhaps it would be useful to set the columns and the row of the tables in a different way, in order to highlight differences between groups more easily. In the conclusion section, I would suggest to include further discussion on the future direction and possible clinical application of the results.

Author Response

Response to comments from reviewers

Reviewer #2

We greatly appreciate the thorough review and kind advice, which has helped us improve the quality of our manuscript. The manuscript has been revised to address the issues raised by the reviewers. The changes made and our point-by-point responses to the comments from the reviewer are summarized below. We have also tracked the changes in red in the revised manuscript.

This is an interesting study that investigated whether nutrition intervention affected food intake after epilepsy surgery and if intravenous infusion was required in patients with epilepsy. Overall, English language is fine; I suggest to check throughout the text for spelling errors and consistent use of abbreviations. The Methods section is clear and provides useful information; inclusion and exclusion criteria, length of follow up, and ethical committee approval have been stated in the text. Primary and secondary outcomes are described adequately. Data in Tables 1 and 2 are quite difficult to read; perhaps it would be useful to set the columns and the row of the tables in a different way, in order to highlight differences between groups more easily. In the conclusion section, I would suggest to include further discussion on the future direction and possible clinical application of the results.

Response:

#1. We are sorry for the errors. We have double checked the entire text for spelling errors and consistent use of abbreviations.

#2. We have changed the Tables in accordance with the advice.

#3. We have added further discussion on the future direction and possible clinical application of the results.
